**EMBO** *reports*

# A non-canonical function of Plk4 in centriolar satellite integrity and ciliogenesis through PCM1 phosphorylation

Akiko Hori[1,2], Karin Barnouin[3], Ambrosius P Snijders[3] & Takashi Toda[1,4,*]

## Abstract

Centrioles are the major constituents of the animal centrosome, in which Plk4 kinase serves as a master regulator of the duplication cycle. Many eukaryotes also contain numerous peripheral particles known as centriolar satellites. While centriolar satellites aid centriole assembly and primary cilium formation, it is unknown whether Plk4 plays any regulatory roles in centriolar satellite integrity. Here we show that Plk4 is a critical determinant of centriolar satellite organisation. Plk4 depletion leads to the dispersion of centriolar satellites and perturbed ciliogenesis. Plk4 interacts with the satellite component PCM1, and its kinase activity is required for phosphorylation of the conserved S372. The nonphosphorylatable PCM1 mutant recapitulates phenotypes of Plk4 depletion, while the phosphomimetic mutant partially rescues the dispersed centriolar satellite patterns and ciliogenesis in cells depleted of PCM1. We show that S372 phosphorylation occurs during the G1 phase of the cell cycle and is important for PCM1 dimerisation and interaction with other satellite components. Our findings reveal that Plk4 is required for centriolar satellite function, which may underlie the ciliogenesis defects caused by Plk4 dysfunction.

**Keywords** centriolar satellites; centrosome; ciliogenesis; PCM1; Plk4
**Subject Categories** Cell Adhesion, Polarity & Cytoskeleton; Post-translational Modifications, Proteolysis & Proteomics

## Introduction

The centrosome consists of two orthogonally arranged centrioles that are surrounded by the pericentriolar material, and plays a multifaceted role in a wide range of biological processes as a major microtubule-organising centre (MTOC) [1]. Importantly, in normal physiological conditions, the centrosome cycle is under strict control regulated by both cell cycle and developmental cues [2]. Plk4 is a conserved protein kinase that executes a pivotal role in the centriole duplication cycle [3]. This kinase is critically a dose-dependent regulator; when malfunctioning, the centrioles fail to duplicate, while when overproduced the centrioles undergo overamplification [4–8]. Plk4 is therefore unanimously regarded as a master kinase for centriole copy number control [3]. In support of this, it has recently been identified that mutations in *Plk4* lead to primordial dwarfism, and abnormal gene amplification results in human embryos exhibiting aneuploidy [9–11].

When starved of serum and/or treated under differentiation signals, cells exit the cell cycle and proceed into the ciliogenesis programme in many cell types [12–14]. Primary cilia are cellular antennas that serve to orchestrate key signalling events required for development. Over the past 10 years, a number of genes encoding centriole/basal body components have been attributed as responsible for a group of human diseases collectively referred to as ciliopathies [13,14]. Despite these advances, our knowledge of the regulatory mechanisms underlying primary cilium formation is far from comprehensive.

Aside from the core architecture of the centrosome, there has been much focus on pericentrosomal structures called the centriolar satellites [15,16]. These non-membranous granules of 70–100 nm in size were originally found through the identification of PCM1 that localises around the centrosome as numerous foci [17,18]. These particles move dynamically towards the centrosome, dependent upon microtubules and the dynein motor. The complete physiological roles of the centriolar satellites have not yet been elucidated; however, at least one of the critical functions is the delivery of centrosomal/ciliary components from the cytoplasm to the centrosome, which aids the formation of the centrosome and primary cilium [17,19–26].

Although Plk4 is essential for centriole duplication, it has not been addressed whether this kinase plays any roles in centriolar satellite integrity. In this study, we have investigated this proposition.

1 The Francis Crick Institute, Lincoln's Inn Fields Laboratory, London, UK
2 Developmental Biomedical Science, Graduate School of Biological Sciences, Nara Institute of Science and Technology (NAIST), Ikoma, Nara, Japan
3 The Francis Crick Institute, Clare Hall Laboratory, London, UK
4 Hiroshima Research Center for Healthy Aging (HiHA), Department of Molecular Biotechnology, Graduate School of Advanced Sciences of Matter, Hiroshima University, Higashi-Hiroshima, Japan
*Corresponding author. Tel: +44 81 082 424 7868; E-mail: takashi-toda@hiroshima-u.ac.jp

We show that Plk4 is required for the spatial distribution and organisation of centriolar satellites separable from its role in centriole duplication. This role is executed by Plk4-dependent phosphorylation of PCM1.

## Results and Discussion

### Plk4 and its kinase activity are required for centriolar satellite integrity

In order to address whether the centrosome/centriole has any impact on centriolar satellite integrity, we first examined the emergence of centriolar satellites under conditions where Plk4 was depleted. Intriguingly, we found that Plk4 knockdown in U2OS cells led to the dispersal of centriolar satellite foci away from the vicinity of the centrosome (detected by an anti-PCM1 antibody) (Fig 1A–C). In line with a previous report [27], Plk4 knockdown resulted in reduced levels of hSAS-6, the procentriolar component essential for centriole duplication [28,29]. However, unlike under Plk4 depletion, hSAS-6 depletion did not display the dispersion of PCM1 (Fig 1B and C). Notably, immunoblotting showed that the total protein levels of PCM1 were not significantly altered upon Plk4 depletion, though a modest reduction was seen (Fig 1A).

Consistent with the notion that PCM1 serves as a structural platform for centriolar satellite assembly [19,30], other components including hMsd1/SSX2IP [23,25,26,31] and BBS4 [32] were simultaneously dispersed upon Plk4 knockdown (Figs 1D and E, and EV1A and B). On the contrary, silencing of Plk4 had no effect on the cytoplasmic localisation of dynein (Fig EV1C). The dispersal of PCM1 upon Plk4 knockdown was not specific to U2OS cells, as the same result was observed in Plk4-depleted HeLa cells (Fig EV1D–F).

In order to address whether satellite dispersion is induced independently of Plk4's role in centriole duplication, U2OS cells were first arrested in G1 phase, followed by Plk4 siRNA treatment (Fig EV1G–I). We found that under this condition, centriolar satellites also became scattered (Fig 1F and G). Hence, the role of Plk4 in centriolar satellite organisation is separable from that of centriole duplication. We then questioned whether Plk4 protein kinase activity was required. U2OS cells were first treated with Plk4-targeting siRNA,

followed by ectopic introduction of RNAi-resistant myc-tagged wild-type (WT*) or kinase-dead Plk4 (KD*). While Plk4-WT* effectively ameliorated the pericentrosomal localisation of PCM1, the introduction of Plk4-KD* was not able to rescue the dispersion of PCM1 (Figs 1H and I, and EV1J and K). Taken together, we conclude that Plk4 kinase activity plays a critical role in the pericentrosomal localisation of centriolar satellites.

### PCM1 physically interacts with Plk4 and is phosphorylated at S372

To test whether PCM1 and Plk4 physically interact within cells, we performed immunoprecipitation. We found ectopically produced EGFP-PCM1 coimmunoprecipitated with endogenous Plk4 (Fig 2A) and exogenous myc-Plk4-bound endogenous PCM1 (Fig 2B). In contrast, immunoprecipitates with an anti-myc antibody (myc-Plk4) did not contain other PCM1-interacting proteins such as hMsd1/SSX2IP [23,25,26,31] and BBS4 [32] (Fig 2A and B). Furthermore, bacterially produced and purified Plk4 and PCM1 also interacted *in vitro* (Fig 2C). These results indicate that the interaction between Plk4 and PCM1 was direct. It is noteworthy that recent proteomics data also identified PCM1 as one of the Plk4-binding proteins [24] and furthermore, that the satellite component Mib1 ubiquitin ligase is involved in proteasome-mediated degradation of Plk4 [33].

We further asked whether PCM1 was phosphorylated and whether Plk4 was able to bind phospho-PCM1. Total cell extracts were prepared from U2OS cells producing myc-tagged Plk4-WT* in which endogenous Plk4 was depleted, and immunoprecipitation was performed with an anti-myc antibody. Pulled-down beads were incubated in the presence or absence of λ-phosphatase and its inhibitors. λ-Phosphatase treatment clearly indicated that PCM1 is a phosphoprotein (Fig EV2A). Collectively, the centriolar satellite component PCM1 is phosphorylated and interacts with Plk4.

Next, we implemented semi-quantitative liquid chromatography–mass spectrometry (LC-MS) to identify the phosphorylation sites within PCM1 that are dependent upon Plk4. Endogenous PCM1 was immunoprecipitated from HeLa cells that were treated with control or Plk4 siRNA. Precipitated PCM1 was cut out from gels (Fig EV2B) and analysed by LC-MS. Inspection of phosphopeptides identified five phospho-sites in the samples treated with control siRNA, of

---

**Figure 1. Plk4 kinase activity is required for the pericentriolar distribution of centriolar satellite components.**

A   Evaluation of siRNA-mediated depletion. U2OS cells were treated with control, Plk4 or hSAS-6 siRNA, and immunoblotting was performed with the indicated antibodies. Asterisk indicates non-specific bands. The positions of molecular weight markers (kDa) are shown on the right.

B   U2OS cells were transfected with the indicated siRNAs. Cell peripheries are marked with dotted lines. Enlarged single cell images (marked with arrowheads in the top row) are shown at the bottom. Scale bars, 5 μm (bottom), 10 μm (top).

C–E   Quantification of the dispersion of satellite components upon the depletion of Plk4 but not hSAS-6 knockdown. The results of PCM1 (C), hMsd1/SSX2IP (D) or BBS4 (E) are shown. Data represent the mean + SD (> 200 cells, *n* = 3). Statistical analysis was performed using two-tailed unpaired Student's *t*-tests. **P < 0.01, n.s. (not significant).

F, G   U2OS cells were arrested in G1 and treated with control or Plk4 siRNA, followed by immunofluorescence microscopy by using indicated antibodies. Enlarged images (marked by arrowheads) are shown on the right. Quantification of PCM1 distribution is shown in (G). > 200 cells were counted and classified into two categories: normal (around and away from the centrosome) or dispersed. Data represent the mean ± SD (*n* = 3). Statistical analysis was performed using two-tailed unpaired Student's *t*-tests. **P < 0.01. Scale bar, 1 μm (F, right), 10 μm (F).

H, I   siRNA-treated U2OS cells were transfected with empty vectors (EV) or plasmids containing Plk4 or siRNA-resistant Plk4-myc WT* or KD* and immunostained with the indicated antibodies. Cell peripheries are marked with solid lines, in which cells with yellow lines are successfully transfected (positive in myc signals), while those with white lines represent non-transfected cells. Quantification of PCM1 distribution is shown in (I). Data represent the mean ± SD (> 200 cells, *n* = 3). Statistical analysis was performed using two-tailed unpaired Student's *t*-tests. ****P < 0.0001, **P < 0.01, n.s. (not significant). Scale bar, 5 μm (H).

Source data are available online for this figure.

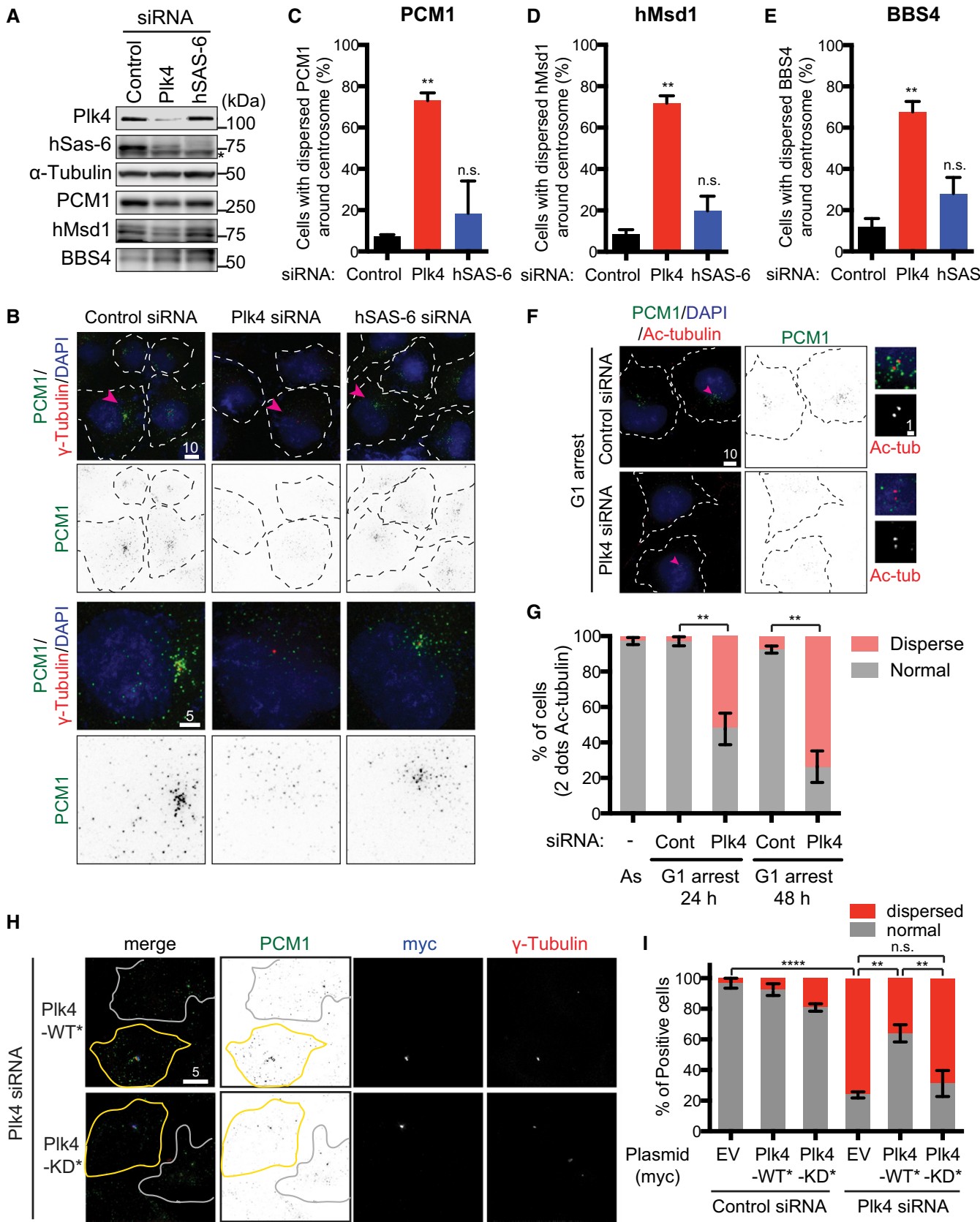

Figure 1.

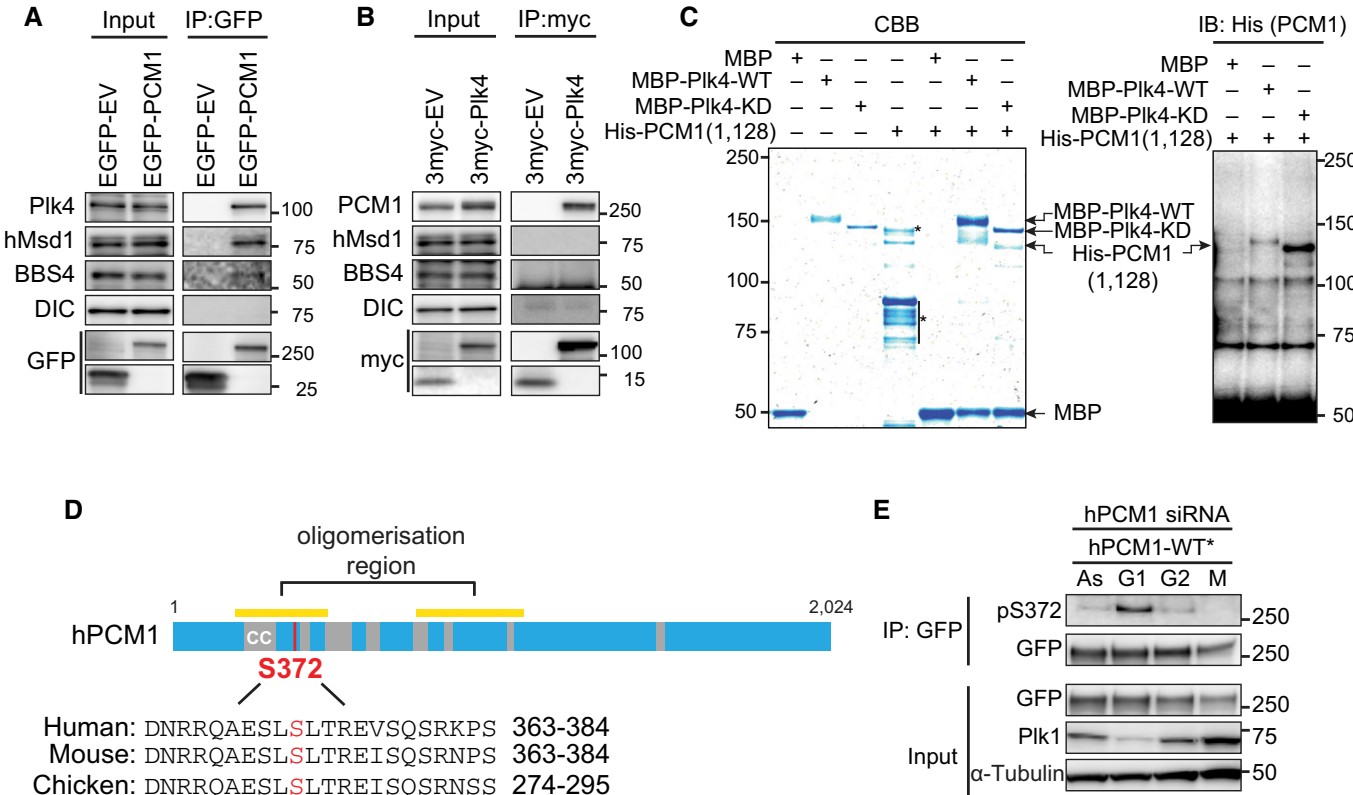

**Figure 2. PCM1 physically interacts with Plk4 and is phosphorylated at S372.**

A, B   Either empty vector plasmids (EV) or those producing EGFP-PCM1 (A) or myc-Plk4 (B) were transfected into U2OS cells. Immunoprecipitation (IP) was performed with anti-GFP (A) or anti-myc antibodies (B), followed by immunoblotting with the indicated antibodies. DIC, the dynein intermediate chain.

C   Bacterially expressed and purified His-PCM1 (a.a. 1–1,128) was incubated with MBP alone or MBP-Plk4 (WT or KD) *in vitro* and pulled down by magnetic beads coupled with an anti-MBP antibody, followed by staining with Coomassie Blue (left) or immunoblotting with an anti-His antibody (right). Asterisks show non-specific bands. The mobility difference between Plk4-WT and Plk4-KD was reported previously [43,48]. We reproducibly observed the upward mobility shift of His-PCM1 upon binding to MBP-Plk4-WT, but not Plk4-KD.

D   Overall structure of PCM1 and the Plk4-mediated phosphorylation site. The conserved serine residue that is phosphorylated in a Plk4-dependent manner is indicated in red.

E   Total cell extracts were prepared from U2OS cells treated with PCM1 siRNA and transfected with RNAi-resistant EGFP-PCM1-WT*. Cells were cultured asynchronously (As) or arrested in G1, G2 or M phase, followed by immunoprecipitation with an anti-GFP antibodies. Pulled-down samples were immunoblotted with the indicated antibodies.

which only one site S372 was hypophosphorylated upon Plk4 knockdown (Fig EV2C–E). S372 is conserved within vertebrate PCM1 homologues and locates in the coiled-coil-rich N-terminal region, although S372 itself is not included within the coiled-coil domain (Fig 2D). It is of note that previous phosphoproteomic studies also identified S372 as one of the *in vivo* phosphorylation sites of PCM1 [34,35].

We constructed EGFP-tagged PCM1 mutants containing non-phosphorylatable (S372A*) or phospho-mimetic versions (S372D* and S372E*) in an siRNA-resistant manner (*) and introduced them into cells in which endogenous PCM1 was depleted. Cells producing PCM1-S372A* or -S372E* contained the comparative levels of PCM1 to WT*, and S372D* was somewhat underproduced (Fig EV2F). Interestingly, the mobility of PCM1-S372A* detected by either anti-GFP or anti-PCM antibodies was slightly faster than that of PCM-WT*, presumably reflecting the *in vivo* phosphorylation of PCM1 at this site (Fig EV2F).

We then prepared a phospho-specific antibody against S372. This antibody recognised phospho-S372 only in samples prepared by pull down (Fig EV2G). Accordingly, we performed immunoprecipitation from the cells blocked at different cell cycle stages. As shown in Fig 2E, we found that S372 phosphorylation peaked during G1 phase. Therefore, Plk4 is responsible for PCM1 phosphorylation at S372.

## PCM1 phosphorylation at S372 is critical for its pericentriolar localisation

We next examined the behaviour of PCM1 phospho-mutants within cells in which endogenous PCM1 was depleted. While wild-type PCM1 localised to the pericentriolar region precisely as the endogenous proteins (see Fig 1B), PCM1-S372A* displayed dispersed patterns similar to wild-type PCM1 proteins upon Plk4 depletion (Fig 3A and B). In stark contrast, PCM1-S372D* or PCM1-S372E* localised to the pericentriolar region, although notably in a more

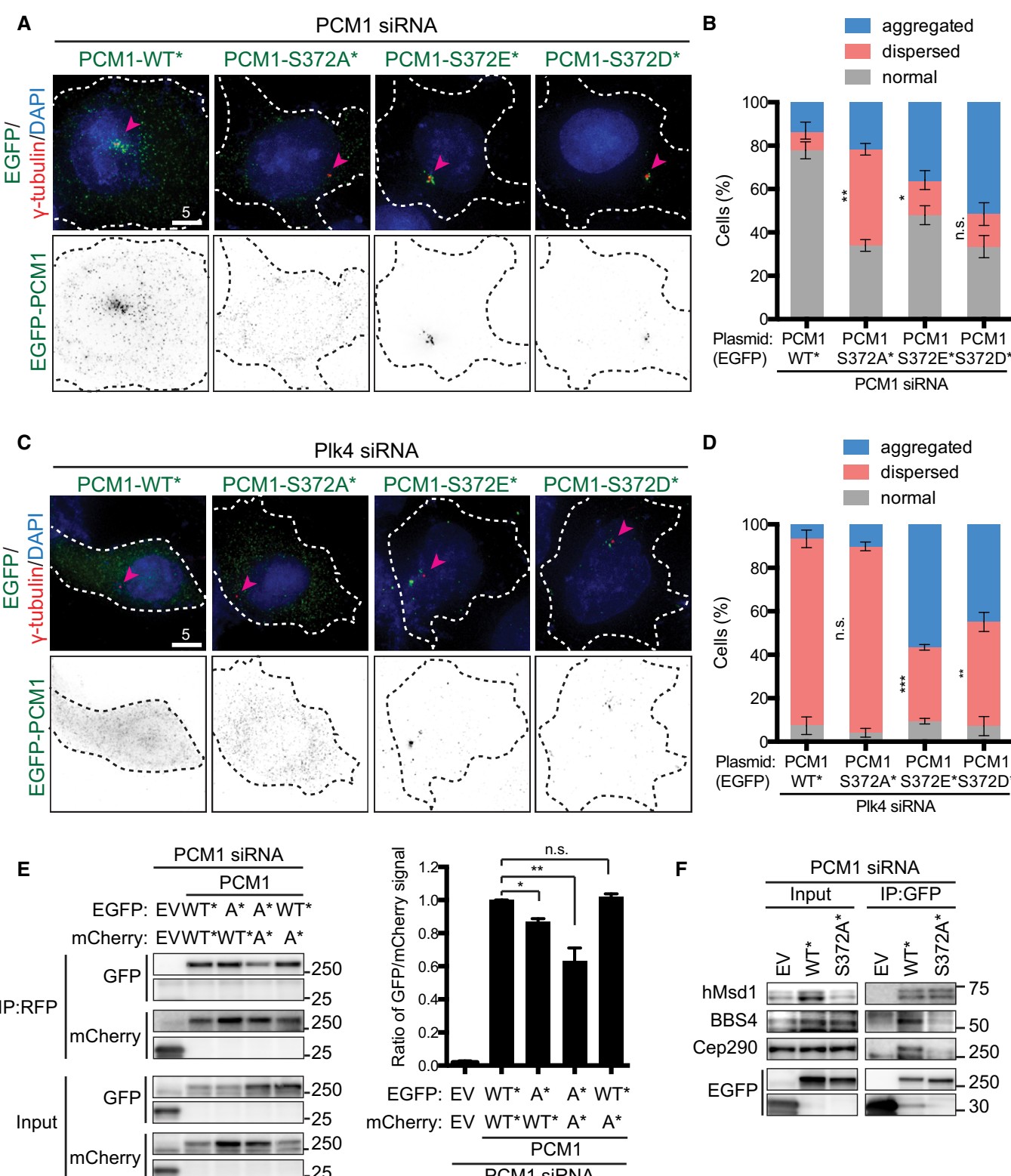

**Figure 3.**

concentrated manner only around the centrosome (classified as "aggregated" in Fig 3B).

We further asked whether the phosphorylation of PCM1 at S372 is responsible for the dispersion of centriolar satellites upon Plk4

depletion. To this end, PCM1-WT*, -S372A*, -S372D* or -S372E* was introduced into U2OS cells depleted of Plk4. While neither PCM1-WT nor PCM1-S372A* was capable of rescuing the dispersed localisation of PCM1, ~50% of cells expressing PCM1-S372D* or

**Figure 3.  Plk4-mediated phosphorylation of PCM1 at S372 is critical for the proper localisation of centriolar satellites, its dimer formation and interaction with other satellite components.**

A–D  siRNA-treated U2OS cells (PCM1 for A and B; Plk4 for C and D) were transfected with plasmids producing various EGFP-PCM1 constructs (WT*, S372A*, S372D* or S372E*), fixed and stained with the indicated antibodies (arrowheads point to the centrosome) (A, C). Quantification data are shown in (B, D). > 200 cells were counted and classified into three categories: normal (around and away from the centrosome), aggregated or dispersed. Data represent the mean ± SD ($n$ = 3). Statistical analysis was performed using two-tailed unpaired Student's $t$-tests. ***$P$ < 0.001, **$P$ < 0.01, *$P$ < 0.05, n.s. (not significant). Note that the percentage of cells displaying "dispersed" is analysed. Scale bars, 5 μm (A, C).

E  PCM1 siRNA-treated U2OS cells were transfected with two types of plasmids (EGFP- and mCherry-tagged) that produce PCM1-WT* and PCM1-S372A*. Immunoprecipitation and subsequent immunoblotting were performed with the indicated antibodies. Quantification data are shown on the right. Data represent the mean + SD ($n$ = 2). Statistical analysis was performed using two-tailed unpaired Student's $t$-tests. *$P$ < 0.05, **$P$ < 0.01, n.s. not significant.

F  U2OS cells were treated with PCM1 siRNA and transfected with empty vectors (EV) or EGFP plasmids that produce PCM1-WT* or PCM1-S372A*. Immunoprecipitation was performed with an anti-GFP antibody, followed by immunoblotting with the indicated antibodies.

PCM1-S372E* showed aggregated appearance in a similar manner to those seen in the presence of Plk4 (Fig 3C and D). We also found that in addition to PCM1, other satellite components including hMsd1/SSX2IP, BBS4 and Cep290 [23,25,26,31,32,36,37] became dispersed in cells producing PCM1-S372A* (Fig EV3A–C). These results established that Plk4 is responsible for the pericentrosomal distribution of centriolar satellites mediated through the phosphorylation of PCM1 at S372. Furthermore, reverse dephosphorylation is likely to be important for the dissolution of these particles.

**S372 phosphorylation is important for dimerisation of PCM1 and its interaction with other satellite components**

We investigated the mechanism by which PCM localisation is regulated through S372 phosphorylation. It is known that PCM1 forms oligomers, which lead to granular PCM1 in cells [19,38]. Furthermore, S372 is situated within the region that is required for self-oligomerisation of PCM1 (201–494) [38] (see Fig 2D). We first asked whether S372 phosphorylation is essential for an intermolecular protein–protein interaction. For this purpose, we constructed plasmids producing differentially tagged PCM1 constructs (EGFP and mCherry) of WT* and S372A* and transfected them in various combinations into cells in which endogenous PCM1 was depleted. Immunoprecipitation with an anti-RFP antibody (which recognises mCherry-PCM1) indicated that mCherry-PCM1-S372A* displayed reduced binding to EGFP-PCM1-WT* and EGFP-PCM1-327A* (Fig 3E). Furthermore, PCM1-S372A* showed compromised interaction with its binding partners such as BBS4 and Cep290, but not hMsd1/SSX2IP (Fig 3F). These results suggested that S372 phosphorylation plays a critical role in PCM1 dimerisation and interaction with other satellite components.

Next, we analysed the motility of PCM1-WT* and PCM1-372E* proteins within the cell. Kymograph analysis of individual PCM1 particles clearly indicated that while PCM1-WT* displayed dynamic motility, PCM1-S372E* exhibited very static trajectories instead (Fig EV3D), and both tracking velocity and length were substantially suppressed in this phospho-mimetic mutant (Fig EV3E–G). Furthermore, other satellite components such as hMsd1/SSX2IP, BBS4 and Cep290 also colocalised with aggregated PCM1-S372E foci (Fig EV3H). Thus, S372 phosphorylation within PCM1 promotes interaction with itself and other satellite components, leading to more aggregation around the centrosome accompanied with reduced motility.

**Plk4-mediated S372 phosphorylation promotes ciliogenesis**

It is known that centriolar satellites play a crucial role in primary cilium formation [15,36,37,39]. We therefore investigated whether Plk4-mediated S372 phosphorylation of PCM1 contributes to ciliogenesis. Using human telomerase-immortalised retinal pigmented epithelial cell lines (hTERT-RPE-1), we first confirmed that upon the depletion of Plk4 but not hSAS-6, PCM1 localised in a diffused manner as in U2OS and HeLa cells (Fig EV4A–C). Furthermore, the depletion of PCM1 or Plk4 impeded efficient ciliogenesis under serum deprivation (Fig 4A and B) as previously reported [30,37,40].

We next examined S372 phosphorylation in serum-starved RPE-1 cells and found that it was phosphorylated (Fig EV4D). We then exogenously produced EGFP-PCM1-WT* or -S372E* in RPE-1 cells upon the depletion of endogenous PCM1. EGFP-PCM1-WT* foci localised as numerous dots in the vicinity of the basal body similar to endogenous PCM1 [36,37] and effectively rescued the ciliogenesis deficiency (Fig 4C and D). EGFP-PCM1-S372E* also rescued the

**Figure 4.  Plk4-mediated phosphorylation of PCM1 at S372 is essential for ciliogenesis.**

A  Immunofluorescence microscopy was performed using the indicated antibodies in hTERT-RPE-1 cells treated with control, PCM1 or Plk4 siRNA. Regions around the basal body (insets) are enlarged on the right-hand side. Scale bars, 5 μm (left) and 1 μm (right).

B  Quantification. At least 150 cells were counted ($n$ = 3). Data represent the mean + SD. Statistical analysis was performed using two-tailed unpaired Student's $t$-tests. ***$P$ < 0.001, ****$P$ < 0.0001.

C  hTERT-RPE-1 cells treated with PCM1 siRNA were transfected by an empty vector (EV) or plasmids producing PCM1-WT*, -S327A* or -S372E*, and immunofluorescence microscopy was performed. Enlarged images of the regions around the basal body are shown in insets. Scale bars, 5 μm and 1 μm (inset).

D  Quantification. At least 200 cells were counted ($n$ = 3). Data represent the mean + SD. Statistical analysis was performed using two-tailed unpaired Student's $t$-tests. *$P$ < 0.05, **$P$ < 0.01, ***$P$ < 0.001, ****$P$ < 0.0001, n.s. not significant.

E  Model. Plk4 plays a decisive role in not only centriole duplication but also centriolar satellite integrity and primary cilium formation. The latter two roles, at least in part, are executed via PCM1 phosphorylation. Plk4 directly or indirectly phosphorylates PCM1 at S372 (I), which promotes its own dimerisation/oligomerisation (II) and interaction with other components of centriolar satellites (III). These processes ensure PCM1 acts as a structural platform for centriolar satellites. Processes II and III are not necessarily sequential steps. It is shown that centriolar satellites and PCM1 play a critical role in centriole duplication and assembly [25,49]. Under conditions of serum starvation or developmental cues, centriolar satellites deliver various factors from the cytoplasm to the basal body, thereby helping the formation of cilia [26,30,37,50,51].

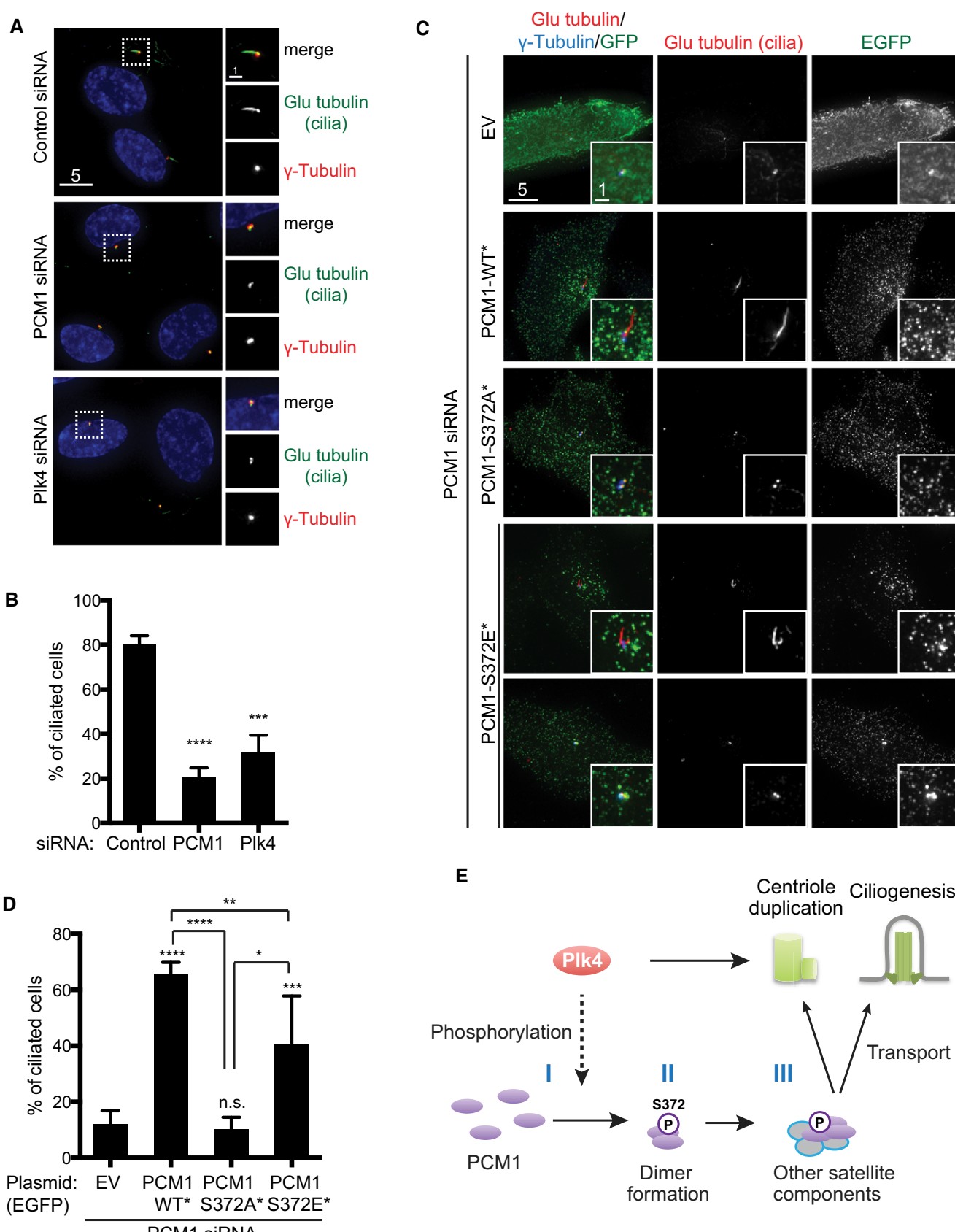

**Figure 4.**

defect, albeit less efficiently compared to wild type (~40% vs. ~65%, Fig 4D) and displayed more concentrated aggregates around the basal body (Fig 4C). Although phospho-mimetic PCM1 appeared capable of rescuing ciliogenesis defects, it might need some reservations. For instance, it is possible that phospho-mimetic PCM1 might interact with residual wild-type PCM1, leading to the formation of functional PCM1 dimers. In clear contrast, however, expression of PCM1-S372A* mutants failed to rescue the ciliogenesis defect and furthermore exhibited the diffused localisation patterns (Fig 4C and D). We also produced PCM1-S372E* upon double depletion of Plk4 and PCM1; under this condition, however, the rescue of ciliogenesis was not readily observed, suggesting that Plk4 may have additional functions in ciliogenesis beyond the phosphorylation of PCM1. In summary, we have uncovered that Plk4-mediated phosphorylation of PCM1 at S372 plays a critical role in primary cilium formation.

### Impact of PCM1 phosphorylation on centriolar satellite integrity

Although it remains to be established whether PCM1 is a direct substrate of Plk4, it is firmly shown that S372 phosphorylation is dependent upon Plk4 kinase activity. We found that PCM1-S372A resulted in the dispersion of PCM1 away from the centrosome. We also showed that this non-phosphorylatable PCM1 could not efficiently form homodimers and displayed compromised binding to BBS4 and Cep290. These findings lead us to propose that S372 phosphorylation plays a critical role in dimer/oligomer formation of PCM1 and interaction with other satellite components, which are prerequisites for its role in centriolar satellite integrity (Fig 4E). At the moment, we could not distinguish whether these two events are independent processes, one process is dependent upon the other, or these are mutually interdependent. In contrast to PCM1-S372A, PCM1-S372D (and PCM1-S372E) formed larger, less motile aggregates around the centrosome. We consider that the dephosphorylation of S372 renders PCM1 particles more motile. It is possible that dephosphorylation is required for destabilising the interaction between PCM1 and the dynein complex [36]. Centriolar satellites are known to dissolve during mitosis and reassemble upon mitotic exit and the following G1 phase [18,19,30]. It is therefore tempting to speculate that this cell cycle-dependent reorganisation of centriolar satellites might be coupled with oscillatory Plk4 activity [41] through PCM1 phosphorylation.

### The role of Plk4 in ciliogenesis

Our study and recent reports in zebrafish, mouse and humans [10,40,42] have shown that Plk4 plays a critical role in ciliogenesis. It is well established that only the mother centriole differentiates into the basal body to assemble the primary cilium [12–14]. If Plk4 is solely responsible for centriole duplication, then how does Plk4 dysfunction lead to cilia defects? Studies using small-molecule inhibitors specific for Plk4 have attributed ciliogenesis defects to the complete loss of centrioles/centrosomes upon prolonged passages of drug-containing cultures [42]. However, studies in zebrafish in which Plk4 was knocked down through morpholino indicated the loss of photoreceptors, which is a characteristic of defective cilia function, as being substantially more penetrant than the disappearance of centrioles (> 90% vs. 25%) [10], and a very similar result was reported in an siRNA-mediated depletion of Plk4 in human cells

[40]. Furthermore, human individuals who carry a homozygous mutation in the *Plk4* gene suffer from ciliopathy-related retinopathy, although patient-derived fibroblasts still contain the centrosome [10]. In line with these reports, we show that ciliogenesis defects were observed in RPE-1 cells upon Plk4 depletion, while the basal body was still retained (Fig 4A and C). It therefore appears that the complete disappearance of the centrosome/centriole may not be an absolute prerequisite of ciliogenesis defects derived from Plk4 dysfunction.

Plk4 is generally thought to be inactive during G1, in which centriole duplication is not initiated while cells undergo ciliogenesis [13,14,41]. It is possible that the requirement of Plk4 for centriolar satellite integrity and ciliogenesis might manifest its temporal function independent of that in centriole duplication. It is, however, of note that recent reports suggest that Plk4 might play a role in centriole duplication during G1 prior to S phase [43,44]. Given these preceding data and our current results, we propose that Plk4 potentiates ciliogenesis to some degree through PCM1 phosphorylation. This safeguards centriolar satellite integrity and promotes delivery of ciliary components to the basal body (Fig 4E). Therefore, Plk4 dysfunction may induce anomalies in humans at least in part, through ciliogenesis failure attributable to centriolar satellite disorganisation.

## Materials and Methods

### Cell cultures

Human cervical cancer HeLa cells and osteosarcoma U2OS cells were cultured in high-glucose DMEM (Invitrogen) supplemented with 10% foetal bovine serum. Immortalised human pigment epithelial cells hTERT-RPE-1 were cultured in DMEM/F12 (Invitrogen) supplemented with 10% foetal bovine serum and 1% non-essential amino acid. HeLa and U2OS cells were cultured in a humidified 5% $CO_2$ incubator at 37°C, and hTERT-RPE-1 cells were cultured in a 10% $CO_2$ incubator at 37°C. HeLa stably expressing GFP-centrin cells (kindly provided by Michel Bornens, Institut Curie, Paris, France) were cultured in DMEM supplemented with 10% foetal bovine serum and 0.5 mg/ml G418. For time-lapse imaging, cells were mounted in L15 media supplemented with 10% FBS.

### RNA interference

Synthetic siRNA oligonucleotides were obtained from Dharmacon-GE Healthcare (Lafayette). The siRNA sequences were 5′-CUGGUA GUACUAGUUCACCUA-3′ (Plk4 siRNA), 5′-GGCUUUAACUAAUUA UGGA-3′ (PCM1 siRNA) or 5′-GCACGUUAAUCAGCUACAAUU-3′ (hSAS-6 siRNA). Control depletion was carried out using siGENOME non-targeting siRNA (Dharmacon). For RNAi experiments, cells were transfected with 40 nM of dsRNA using Lipofectamine RNAi-MAX (Invitrogen), and cells were fixed 48 h after siRNA treatment unless otherwise stated.

### Plasmid construction and DNA transfection

pEGFP-hPCM1 was a gift from Song-Hai Shi (Memorial Sloan Kettering Cancer Center, NY, USA). pcDNA-myc-Plk4 was a gift from

Erich Nigg (Biozentrum, University of Basel, Switzerland). pmCherry-α-tubulin was obtained from Addgene. Plasmids containing various forms of PCM1 or Plk4 were constructed with appropriate PCR primer pairs. The amplified products were each subcloned into pEGFP-C1 and pmCherry-C1 (Clontech). pMBP-Plk4 (WT and KD) and p6His-PCM1 (containing amino acids 1–1,128) were constructed in pMAL (New England BioLabs) and pETM-6His (a gift from Thomas Surrey), respectively. For the construction of RNAi-resistant versions, we introduced 4 or 6 silent substitutions within the PCM1 or Plk4 siRNA-target region. The PCM1 siRNA-target region, 5′-GGCTTTAACTAATTATGGA-3′, was changed to 5′-AGC CCTGACTAATTATGGA-3′ using site-directed mutagenesis with the primers, 5′-ctccacttccataattagtcagggcttgagtttcagctttagacaaattaatagaca ctggac-3′ and 5-gtccagtgtctattaatttgtctaaagctgaaactcaagccctgactaattat ggaagtggag-3′. The Plk4 siRNA-target region, 5′-CTGGTAGTACTAG TTCACCTA-3′, was changed to 5-CTGGCAGCACGAGCTCTCCAA-3′ using site-directed mutagenesis with the primers, 5′-caataatcatagg aagaaaacctggcagcacgagctctccaaaggccttatcacctcctccttctg-3′ and 5-cag aaggaggaggtgataaggcctttggagagctcgtgctgccaggtttctttcctatgattattg-3′. RNA-resistant constructs were denoted by an asterisk (*, i.e. Plk4-WT*). Plk4-KD* (K41R/D154A) was made by site-directed mutagenesis (QuikChange Lightning). Serine 372 within PCM1 was further mutated to alanine (S372A), aspartate (S372D) or glutamate (S372E).

Cells were treated with siRNAs for 48 h and observed under the microscope. For double transfection experiments, cells were treated with siRNAs for 48 h, followed by the second transfection with various plasmids. Cells were observed under the microscope 24 h later.

## Cell cycle synchronisation

For G1 cell cycle arrest, U2OS cells were doubly treated with mimosine (0.2 mM; Sigma) for 16 h each prior to further siRNA treatment. Immunofluorescence microscopy and immunoblotting were performed 24 h and 48 h later. For G2- and M-phase arrest, U2OS cells were treated for 24 h with RO-3306 (1 μM; ENZO Life Sciences) and nocodazole (50 ng/ml; Sigma), respectively.

## Recombinant protein expression and purification

Plasmids producing MBP, MBP-Plk4-WT or MBP-Plk4-KD were introduced into *E. coli* strain BL21-CodonPlus(DE3)-RIL (Agilent Technologies), and protein expression was induced by adding 0.05 mM IPTG at 18°C for 16 h. Subsequently, proteins were purified on amylose resin (New England Biolabs) according to the manufacturer's instructions and concentrated by Vivaspin 20 (GE Healthcare). Purified proteins were stored in MBP buffer (20 mM Tris–HCl pH 7.5, 100 mM NaCl, 1 mM DTT, 10% glycerol, EDTA-free complete protease inhibitor cocktail (Roche)). The recombinant His-PCM1(1128) protein was produced essentially in the same manner as above. Proteins were purified on complete His-Tag Purification resin (Roche) according to the manufacturer's instructions. Fusion protein was eluted with elution buffer (20 mM Tris–HCl pH 7.5, 100 mM NaCl, 300 mM imidazole, 1 mM DTT, 10% glycerol, EDTA-free complete protease inhibitor cocktail). Eluted protein was desalted into a final buffer (20 mM Tris–HCl pH 7.5, 100 mM NaCl, 1 mM DTT, 10% glycerol) using PD-10 desalting columns (GE Healthcare) and concentrated by Vivaspin 20.

## *In vitro* binding assay

For *in vitro* binding assay, MBP, MBP-Plk4-WT or MBP-Plk4-KD was mixed with Dynabeads Protein G (Invitrogen) coupled with an anti-MBP antibody; 8 μg of His-PCM1(1128) was treated with 600 μl binding buffer (50 mM Tris–HCl pH 7.5, 200 mM NaCl, 1 mM DTT, 2 mM MgCl$_2$, 0.5 mg/ml BSA, 10% glycerol, 0.05% NP-40, EDTA-free complete protease inhibitor cocktail) and incubated at 4°C for 2 h. After several washes with the binding buffer (without BSA), beads were boiled in Laemmli sample buffer and applied to SDS–PAGE. Gels were stained with Coomassie Brilliant Blue or immunoblotted using antibodies against MBP, His or phospho-S372.

## Antibodies

For immunofluorescence microscopy, the following antibodies were used: chicken anti-GFP (1:300, ab13970; Abcam), chicken anti-myc (1:300, A-21281; Molecular Probes-Thermo Fisher Scientific), rabbit anti-SSX2IP (1:150, HPA027306; Sigma-Aldrich), rabbit anti-PCM1 (1:300, sc67204; Santa Cruz Biotechnology), rabbit anti-γ-tubulin (1:250, T5192; Sigma-Aldrich), mouse anti-α-tubulin (1:250, T9026, Sigma-Aldrich), mouse anti-Plk4 (1:200, H00010733-B01; Abnova), mouse anti-Glu-tubulin (1:150, AB3201; Merck Millipore), anti-DIC (1:150, MAB1618; Merck Millipore), anti-BBS4 (1:100, Abnova), anti-MBP (1:10,000, New England Biolabs) and anti-His antibodies (1:2,000, Novagen). Secondary antibodies were Alexa Fluor 488-coupled anti-rabbit, Alexa Fluor 594-coupled anti-rabbit, Alexa Fluor 594-coupled anti-mouse, Alexa Fluor 488-coupled anti-mouse, Alexa Fluor 488-coupled anti-chicken or Cy3-coupled anti-mouse antibodies (all used at 1:1,500, Molecular Probes). For immunoblotting, the following antibodies were used: mouse anti-GFP (1:1,000, 11814460001; Roche), mouse anti-Cherry (1:1,000, 632543; Clontech), mouse anti-myc (1:1,000, MMS-150R; Babco), rabbit anti-hSAS-6 (1:1,000, sc98506; Santa Cruz Biotechnology), rabbit anti-PCM1 (1:2,000), mouse anti-Plk4 (1:1,000), rabbit anti-SSX2IP (1:1,000), anti-DIC (1:1,000), anti-BBS4 (1:1,000), rabbit anti-γ-tubulin and mouse anti-α-tubulin antibodies (1:5,000).

Rabbit polyclonal phospho-specific antibody raised against phosphopeptide (CQAESLS$^{PO3H2}$LTREVS) was generated and affinity-purified by Eurogentec. This antibody recognises phospho-S372 only when PCM1 is pulled down from the whole-cell extracts; we could not detect phosphorylated PCM1 with simple immunoblotting against the total protein lysates or immunofluorescence microscopy.

## LC-MS analysis

1.5 mg of total protein extracts was prepared from HeLa cell cultures treated with control or Plk4 siRNA, and immunoprecipitation was performed with an anti-PCM1 antibody. Colloidal Coomassie-stained bands corresponding to precipitated PCM1 were cut out from gels and subject to trypsin digestion [45] and Q Exactive LC-MS analysis (Thermo Fisher Scientific). The data were searched against human Uniprot (UniProt KB2012_08 taxonomy human 9606 canonical with contaminants 20120921) using the Andromeda search engine and MaxQuant (version 1.3.0.5) [46], as well as Mascot Daemon search engines (version 2.4.0, Matrix Science). For MaxQuant, a false discovery rate of 0.1% was used to generate protein and peptide identification tables.

The data were uploaded into Perseus (MaxQuant) for further statistical analyses. Skyline software was employed for label-free quantification (version 3.1) [47]. The extracted ion chromatograms were manually curated to ensure accurate peak picking and quantification.

**Immunofluorescence microscopy and image analysis**

Immunofluorescence microscopy was performed as described previously [25,26]. Briefly, the cells were fixed with methanol at −20°C for 5 min and washed in PBS. After blocking in 3% BSA for 1 h at room temperature, cells were incubated with primary and then secondary antibodies. DNA was visualised by the addition of DAPI (4,6-diamidino-2-phenylindole; Vector Lab). During time-lapse imaging, the cells were kept at 34–37°C by a chamber heater.

Images were taken using an Olympus IX71 wide-field inverted epifluorescence microscope with Olympus PlanApo 60×, NA 1.4, or UApo 40×, NA 1.35, oil immersion objectives (Olympus). DeltaVision image acquisition software (softWoRx 3.3.0; Applied Precision Co.) equipped with Coolsnap-HQ digital CCD camera or Cascade EMCCD 512B camera (Roper Scientific) was used. The sections of images were compressed into a two-dimensional (2D) projection using the DeltaVision maximum intensity algorithm. Deconvolution was applied before generating the 2D projection. Images were taken as 64 sections along the *z*-axis at 0.2-μm intervals. Captured images were processed with Adobe Photoshop CS3 (version 10.0).

The localisation patterns of centriolar satellite components were categorised by visual inspection. When multiple immunofluorescence signals of satellite components were found at the centrosome region (marked by anti-γ-tubulin dots), they were categorised as "normal". In contrast, when these signals were dispersed away from the centrosome, they were classified as "dispersed". When signals formed larger aggregates, they were scored as "aggregated".

**Immunoprecipitation**

For coimmunoprecipitation, 1 mg of cell lysate was incubated with 30 μl GFP-Trap or RFP-Trap (ChromoTek) in lysis buffer (25 mM Tris–HCl, pH 7.0, 1 mM EDTA, 300 mM NaCl, 10% glycerol, 1% NP-40, 1 mM DTT, 10 mM NaF, 25 mM DMSF and EDTA-free protease inhibitor tablet (Complete: Roche)) overnight at 4°C. After washing with lysis buffer, the beads were denatured at 95°C in NuPAGE buffer (Invitrogen) and run on SDS–PAGE, followed by immunoblotting.

**Quantification and fluorescence signal intensity measurement**

For fluorescence signal intensity measurement, fluorescence signals were quantified using maximum intensity, after subtracting background signals in the vicinity of the fluorescent spot. The SoftWoRx software was used for analysis. At least 200 cells were counted in each sample, independently, three times, from which standard deviations and *P*-values were calculated.

**Tracking the trajectory of PCM1 motility**

U2OS cells were treated with PCM1 siRNA for 48 h and then further transfected with pmCherry-α-tubulin and pEGFP-PCM1-WT* or pEGFP-PCM1-S372E* plasmids. 18 h after the second transfection, live imaging was started. Movie streams were acquired at a frame rate of 2 frame/s. Subsequent particle tracking analysis was carried out using Imaris software (Bitplane). Briefly, the diffusion trajectories of single particles were determined by connecting the spots corresponding to individual time points.

**Statistical data analysis**

All data represent the mean of multiple experiments ± SD. Experiment sample numbers and the number of replicates used for statistical testing have been reported in the corresponding figure legends. All *P*-values are from two-tailed unpaired Student's *t*-tests. Unless otherwise stated, we followed this key for asterisk placeholders for *P*-values in the figures: ****$P < 0.0001$, ***$P < 0.001$, **$P < 0.01$, *$P < 0.05$.

**Expanded View** for this article is available online.

## Acknowledgements

We thank Michel Bornens, Erich Nigg, Wei Shao, Song-Hai Shi and Thomas Surrey for reagents. We thank Yasuto Murayama for reagents and technical advice. We are grateful to Risa Mori for critical reading of the manuscript. T.T. and A.P.S were supported by Cancer Research UK.

## Author contributions

AH and TT designed the experiments; AH performed the majority of the experiments and data analysis; and KB performed LC-MS and together with APS conducted comparative analysis of phosphopeptides; AH and TT wrote the paper with suggestions from KB and APS.

## Conflict of interest

The authors declare that they have no conflict of interest.

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
