## [Review Process File · EMBO Reports]

Manuscript EMBO-2015-41432

A non-canonical function of Plk4 in centriolar satellite integrity and ciliogenesis through PCM1 phosphorylation

Akiko Hori, Karin Barnouin, Ambrosius S. Snijders, and Takashi Toda

Corresponding author: Takashi Toda, The Francis Crick Institute

Review timeline:

Submission date:	16 June 2015
Editorial Decision:	31 July 2015
Resubmission:	22 September 2015
Editorial Decision:	09 November 2015
Revision received:	11 November 2015
Editorial Decision:	23 November 2015
Revision received:	24 November 2015
Accepted:	30 November 2015

Editor: Esther Schnapp

Transaction Report:

1st Editorial Decision

31 July 2015

Thank you for the submission of your manuscript to our journal. I apologize for the unusual delay in getting back to you; we have only today received the third report on your manuscript. The full set of reports is copied below.

I am sorry to say that neither referee strongly supports publication of the study by EMBO reports. As you will see, both referees 1 and 3 point out that it needs to be shown whether Plk4 depletion results in the loss of centrioles, which might affect satellite dispersal indirectly. All referees also raise concerns regarding the phosphorylation of PCM1 by Plk4, whether the interaction of both proteins is direct or indirect, whether it depends on satellites, and whether S372 is phosphorylated in vivo and after mitotic exit. Referee 2 adds that endogenous proteins and the interaction of PCM1 with satellite proteins should be investigated.

Given these concerns, the amount of work required to address them, the uncertain outcome of these experiments, and the fact that EMBO reports can only invite revision of papers that receive enthusiastic support from a majority of referees, I am sorry to say that we cannot offer to publish your manuscript at this point.

However, in case you feel that you can fully address the referee concerns (as mentioned above and in their reports) in a timely manner and obtain data that would considerably strengthen the message of the study, then we would have no objection to consider a new manuscript on the same topic in the near future. Please note that if you were to send a new manuscript this would be treated as a new

submission rather than a revision and would be reviewed afresh, also with respect to the literature and the novelty of your findings at the time of resubmission.

At this stage of analysis, I am sorry to have to disappoint you. I nevertheless hope, that the referee comments will be helpful in your continued work in this area, and I thank you once more for your interest in our journal.

REFEREE REPORTS

Referee #1:

This study claims that phosphorylation of PCM1 at S372 by Plk4 is required for its dimerisation to form a platform necessary for ciliogenesis. While there are elements of great interest in these findings - specifically the identification of the S372 phosphorylation site which is well documented - the conclusions drawn go beyond the experimental evidence provided.

One difficulty in interpreting these findings is to know the status of the centrioles in specific Plk4 depleted cells - is the daughter centriole present; has the process of centriole to centrosome conversion occurred on residual centrioles; what is the phase of the cell cycle; how much Plk4 has been depleted? All of these factors might influence the proposed failures of the mother centriole to organize centriolar satellites.

In the second part of the study, it is claimed that Plk4 and PCM1 physically interact. Whilst convincing evidence is presented for their co-immuno-precipitation, these findings fall short of demonstrating a direct physical interaction between these molecules. As it is, the relevance of the co-IP is not clear.

A similar difficulty is also apparent in the interpretation of the experiments in Figure 3. These are all couched in terms of formation of PCM1 dimers. The interactions described give no indication of stoichiometry. They show the differentially tagged molecules interact not that they form dimers. Finally, all microscopy has been carried out at a low level of resolution. This is surprising since the study concerns centrioles and cilia, structures that can be analysed through super-resolution light microscopy and electron microscopy. Thus it becomes impossible to know exactly what is happening to the centriole, its satellites or the cilia.

Referee #2:

The present manuscript of Hori et al. addresses the idea of the regulation of centriolar satellite formation by Plk4. The authors show evidence, using RNAi, that knockdown of Plk4 affects satellite accumulation in the vicinity of centrosomes. They further identify a phosphorylation site in PCM1, which may be phosphorylated by Plk4. Single point mutants of this site, abolishing the phosphorylation, interfere with localization of the respective variants in satellites. The authors show evidence that they possibly compromise homotypic dimerization of PCM1 and finally do not support ciliogenesis.

The paper contains significant novel data, which are of general cell-biological interest. The quality of the data is, as usually, excellent. I still have some problems with the interpretation of the data and the way conclusions are drawn on the basis of the experiments presented here. These drawbacks of the manuscripts prevent me from supporting publication in EMBO Rep at this stage.

Major issues:

The authors use their GFP-tagged PCM1 variants throughout the manuscript and draw conclusions about the overall satellite structure under these conditions. I think this is not correct: in several experiments, the exogenous GFP-tagged protein is the only marker they look at. No further analysis determines how the endogenous satellite proteins behave under these conditions. For instance: The authors state that "This result firmly established that Plk4 is responsible for the pericentrosomal distribution of centriolar satellites mediated through phosphorylation of PCM1 at

S372." (p.8/top). What the results shows is that that S372 phosphorylation regulates integration of exogenous PCM1 into centriolar satellites. Given that PCM1 is the platform for satellite assembly, this may indicate dispersal of satellites when this phosphorylation is lost. However, no endogenous satellite protein is detected in this experiment. Looking to at least one additional component, e.g. Msd1 or Cep290, after replacement of PCM1 by the phospho-mutant is certainly required to clarify that issue.

Along the same lines: the differences in oligomerisation behavior are detectable but rather weak, especially considering the fact that there are no statistics. What about interaction of PCM1 with all the other satellite proteins? If this is really the way satellite structure as a complete entity is regulated, it should be investigated how the interaction of PCM1 with e.g. BBS4, Msd1, Cep290 is affected by phosphorylation. The authors even mention the interaction with Cep290 as a regulated event in their discussion.

The tracking data are very interesting. However, the same applies to these experiments as to the one above: Does expression of the mutants, or, ideally replacement of endogenous PCM1 by the variants, change the collective behavior of satellite proteins?

The experiments on primary cilia formation are interesting as well. They suggest that the phosphorylation is maintained upon mitotic exit, i.e. upon degradation of a significant part of Plk4. I agree that the described genetic data on ciliogenesis defects after loss of Plk4 argue for this possibility. However, I think it needs to be shown that the phosphorylation is indeed still preserved in cells after mitotic exit. Moreover, it is very unclear how the defect arises. There seems to be no axoneme extension. Why is that? There is no explanation provided, which would give us an idea about the mechanism of phosphorylation regulation on primary cilia formation.

RNAi mediated knockdown of Plk4 reduces S372 phosphorylation of PCM1. This could be a direct effect but certainly also indirect. Evidence from a reconstitution of the phosphorylation with purified components is missing. It is therefore an overstatement to claim that a direct phosphorylation is shown.

Minor points:

The way quantifications were done is not entirely clear to me. If I understand it correctly, cells were visualized by eye and categorized. It should be briefly mentioned how it was done.

"S372 is conserved within vertebrate Plk4 homologues and locates in the coiled coil rich N-terminal region, although S372 itself is not included in the coiled-coil domain (Fig 2D)" (p.6/7).... Should be: ... vertebrate PCM1...

"Interestingly, the mobility of Plk4-S372A* detected by either anti-GFP or anti-PCM antibodies was slightly faster than that of Plk4-WT*, presumably reflecting the in vivo phosphorylation of Plk4 at this site (Fig 2E)" (p 7) should be corrected to PCM1 where obviously Plk4 is wrong

This sounds like a paradox to me: "However, the difference of PCM1 mobility was hardly distinguishable between wild type and kinase-dead Plk4 containing cells. Collectively, the centriolar satellite component PCM1 is phosphorylated and interacts with Plk4."

Fig. 3B: very few microtubules are seen on the image of the S372 variant tracking. Is that significant, i.e. could the phosphorylation also affect microtubule binding?

The legend of the model now tells us: Whether Plk4 directly phosphorylates PCM1 and where this phosphorylation occurs are currently unknown.

Referee #3:

This work identifies a connection between Plk4, a kinase required for centriole duplication, and PCM1, the protein thought to be the scaffold of centriolar satellites. Satellites were first described many years ago, but there is currently much interest in them because they are connected in molecularly mysterious ways to many functions of centrosomes and cilia. Here the authors find that depletion of Plk4 causes a change in the distribution of satellites, and that Plk4 and PCM1 have a physical interaction in lysates. They also identify and characterize phosphorylation sites on PCM1. This putative connection is interesting, but there are some fundamental problems with the paper that make it difficult to be enthusiastic about publication.

1) The phenotypes with respect to satellites are assayed by depleting Plk4 by RNAi. Does the depletion of Plk4 result in loss of centrioles? It should, by published accounts. If so, then the dispersal of satellites is likely to be due to lack of a focused microtubule aster, or possibly lack of centrosomally-anchored microtubules (to the sub-distal appendages, for example). These would be much better experiments if done with centrinone, the new inhibitor of Plk4 described in a paper from Oegema and colleagues. This would allow acute inhibition of Plk4, without loss of centrioles, which requires passage through mitosis, and thus assessment of a Plk4 function in satellites without the confounding effect of loss of centrioles.

2) The putative interaction of Plk4 with PCM1. Is this occurring in the context of the satellite structure, or with the two proteins independently of satellites? The lack of other satellite proteins in the Plk4 IP suggests the latter. What does that mean then? The authors cite another paper (ref. 25) as also identifying a Plk4-PCM1 interaction, but that paper seems to only be about proximity biotin labeling, which says something about the proteins being in the same vicinity. Helpful, but not evidence for a physical interaction.

3) PCM1 phosphorylation. PCM1 has many S/T sites that have been identified as phosphosites in large-scale efforts (see http://www.hprd.org/ptms?hprd_id=02624&isoform_id=02624_2&isoform_name=Isoform_1). It would be helpful to know whether S372 is actually phosphorylated in vivo or not, and whether this depends on Plk4. It could be that the reason the migration of the protein does not appreciably change in the Plk4 kinase dead is that there are many other phosphorylation sites that are unaffected.

Resubmission - authors' response

22 September 2015

We would like to submit our manuscript entitled "Plk4 regulates centriolar satellite integrity and promotes ciliogenesis through PCM1 phosphorylation" by Hori et al. to EMBO Reports as a Scientific Report. This is a new submission that was derived from our previous submission (EMBOR-2015-40866V1).

In the decision letter, you wrote "*in case you feel that you can fully address the referee concerns (as mentioned above and in their reports) in a timely manner and obtain data that would considerably strengthen the message of the study, then we would have no objection to consider a new manuscript on the same topic in the near future.*"

As described below in point-by-point details, we addressed all the points raised by you and the three referees. We believe that this new manuscript deserves consideration for publication in EMBO Reports. The manuscript consists of 4 figures and Supplementary Information (Supplementary materials and methods, 4 figures and references). The number of characters is 23,869 (excluding references).

Both referees 1 and 3 point out that *it needs to be shown whether Plk4 depletion results in the loss of centrioles, which might affect satellite dispersal indirectly*. All referees also raise concerns regarding the phosphorylation of PCM1 by Plk4, whether the interaction of both proteins is direct or indirect, whether it depends on satellites, and whether S372 is phosphorylated in vivo and after mitotic exit. Referee 2 adds that *endogenous proteins and the interaction of PCM1 with satellite proteins should be investigated*.

To distinguish the centriolar satellite dispersion phenotype from centriole duplication defects, we arrested cells in G1 and then treated cells with Plk4 siRNA, by which Plk4 depletion did/could not lead to centriole duplication defects. As shown in new **Figure 1F and G**, under this condition, satellite dispersion still occurred.

Regarding the interaction between Plk4 and PCM1, we performed an in vitro binding assay by using bacterially produced and purified proteins. They interacted (new **Figure 2C**).

With regards to whether S372 is phosphorylated in vivo upon mitotic exit, we prepared an S372-phospho specific antibody. Although this antibody worked only for IP samples, but not for simple immunoblotting, it was useful to address the point raised regarding in vivo phosphorylation and the cell cycle stage. We present that phosphorylation occurred during G1 upon mitotic exit (new **Figure 2E**).

We also showed that in cells producing non-phosphorylatable PCM1-S372A, the interaction between PCM1-S372 and other satellites components (BBS4 and Cep290), but not hMsd1/SSX2IP, were impaired (new **Figure 3F**).

Referee #1

We appreciate this referee's critical and thoughtful review.

One difficulty in interpreting these findings is to know the status of the centrioles in specific Plk4 depleted cells - is the daughter centriole present; has the process of centriole to centrosome conversion occurred on residual centrioles; what is the phase of the cell cycle; how much Plk4 has been depleted? All of these factors might influence the proposed failures of the mother centriole to organize centriolar satellites.

the daughter centriole present

Staining with an anti-acetylated tubulin antibody showed that daughter centrioles exist in G1-arrested U2OS cells in which Plk4 was depleted, leading to satellite dispersion (**Figure 1F**, two dots shown in insets).

what is the phase of the cell cycle

As described earlier, to distinguish the well-established centriole duplication defects from the centriolar satellite dispersion phenotype upon Plk4 depletion, we first arrested cells in G1 and then treated cells with Plk4 siRNA, by which Plk4 depletion did/could not lead to centriole duplication defects. Under this condition, satellite dispersion still occurred (**Figure 1F and G**). Furthermore, by using newly prepared S372-phospho specific antibody, we showed that S372 phosphorylation occurred in G1 (**Figure 2E**).

how much Plk4 has been depleted

The degree of Plk4 levels upon siRNA treatment was shown in **Figure 1A** and Supplementary **Figure S1H and S1I**.

In the second part of the study, it is claimed that Plk4 and PCM1 physically interact. Whilst convincing evidence is presented for their co-immuno-precipitation, these findings fall short of demonstrating a direct physical interaction between these molecules. As it is, the relevance of the co-IP is not clear.

We expressed and purified Plk4 and PCM1 proteins in bacteria, and showed that these two proteins interact in vitro (**Figure 2C**).

A similar difficulty is also apparent in the interpretation of the experiments in Figure 3. These are all couched in terms of formation of PCM1 dimers. The interactions described give no indication of stoichiometry. They show the differentially tagged molecules interact not that they form dimers.

We agree that from this experiment (Figure 3E), we could not discuss stoichiometry. We can only argue that S327 phosphorylation is important for self-dimerisation. We believe that it is not irrational to argue that compromised dimerisation would lead to defective oligomerisation of PCM1, which is critical for centriolar satellite organisation.

Finally, all microscopy has been carried out at a low level of resolution. This is surprising since the study concerns centrioles and cilia, structures that can be analysed through super-resolution light microscopy and electron microscopy. Thus it becomes impossible to know exactly what is happening to the centriole, its satellites or the cilia.

We believe that the quality of images presented in this study is satisfactory to show centriolar satellite organisation (around the centrosome or dispersed) and to clarify whether or not centriole, the basal body and cilia are present. In fact, referee 2 praised the quality of our images. EM analysis

would be relevant if we would like to describe the details of defective ciliogenesis but we feel that it is beyond the scope of this current study.

Referee #2

We thank this referee for his/her encouraging comments on our work and several suggestions and requests.

Major issues:

The authors use their GFP-tagged PCM1 variants throughout the manuscript and draw conclusions about the overall satellite structure under these conditions. I think this is not correct: in several experiments, the exogenous GFP-tagged protein is the only marker they look at. No further analysis determines how the endogenous satellite proteins behave under these conditions.

For instance: The authors state that "This result firmly established that Plk4 is responsible for the pericentrosomal distribution of centriolar satellites mediated through phosphorylation of PCM1 at S372." (p.8/top). What the results show is that S372 phosphorylation regulates integration of exogenous PCM1 into centriolar satellites. Given that PCM1 is the platform for satellite assembly, this may indicate dispersal of satellites when this phosphorylation is lost. However, no endogenous satellite protein is detected in this experiment. Looking to at least one additional component, e.g. Msd1 or Cep290, after replacement of PCM1 by the phospho-mutant is certainly required to clarify that issue.

In response to this comment, we performed immunofluorescence microscopy using anti-hMsd1, anti-BBS4 and anti-Cep290 antibodies to examine the localisation of these endogenous satellite components upon introduction of non-phosphorylatable PCM1 (PCM1-S372A). As shown in **Supplementary Figure S3A-C**, all these satellite components were also dispersed in PCM1-S372A producing cells. In the case of PCM1-S372E, all these three components colocalised with aggregated PCM1-S372E around the centrosome (**Supplementary Figure S3H**).

Along the same lines: the differences in oligomerisation behavior are detectable but rather weak, especially considering the fact that there are no statistics. What about interaction of PCM1 with all the other satellite proteins? If this is really the way satellite structure as a complete entity is regulated, it should be investigated how the interaction of PCM1 with e.g. BBS4, Msd1, Cep290 is affected by phosphorylation. The authors even mention the interaction with Cep290 as a regulated event in their discussion.

This is an important point. Thank you very much for pointing this out. We performed IP between PCM1-S372A and other satellite components including BBS4, hMsd1 and Cep290. As shown in **Figure 3F**, interaction with BBS4 and Cep290 were impaired, while that with hMsd1 was not. This indicates that S372 phosphorylation is important for not only dimerisation but also for the interaction with other satellite components. This notion is described in the text (page 8) and a proposed model (**Figure 4E**) incorporates this point.

The tracking data are very interesting. However, the same applies to these experiments as to the one above: Does expression of the mutants, or, ideally replacement of endogenous PCM1 by the variants, change the collective behavior of satellite proteins?

In response to this comment, we examined localisation patterns of other satellite components including Msd1, BBS4 and Cep290 in PCM1-S372E producing cells in which endogenous PCM1 was depleted. As shown in **Supplementary Figure S3H**, all the three components colocalised with PCM1-S372E as aggregated foci around the centrosome.

The experiments on primary cilia formation are interesting as well. They suggest that the phosphorylation is maintained upon mitotic exit, i.e. upon degradation of a significant part of Plk4. I agree that the described genetic data on ciliogenesis defects after loss of Plk4 argue for this possibility. However, I think it needs to be shown that the phosphorylation is indeed still preserved in cells after mitotic exit. Moreover, it is very unclear how the defect arises. There seems to be no axoneme extension. Why is that? There is no explanation provided, which would give us an idea about the mechanism of phosphorylation regulation on primary cilia formation.

We generated S372-phospho specific antibody. Using this tool, we showed that in serum-starved RPE-1 cells, S372 is phosphorylated (**Supplementary Figure S4D**).

With regards to the ciliogenesis defects, we previously published a paper showing that centriolar satellite integrity is critical for transport of ciliary proteins to the basal body via microtubules (Hori et al., 2014). Another group also published the similar results (Klinger et al., 2014). This notion is described and shown in the model (**Figure 4E and the figure legend**).

RNAi mediated knockdown of Plk4 reduces S372 phosphorylation of PCM1. This could be a direct effect but certainly also indirect. Evidence from a reconstitution of the phosphorylation with purified components is missing. It is therefore an overstatement to claim that a direct phosphorylation is shown.

In this new submission, we showed a direct in vitro interaction between Plk4 and PCM1 (**Figure 2C**). Nonetheless, we agree that it is an overstatement to claim that a direct phosphorylation is shown. We toned down the description in the text (page 9, "Although it remains to be established whether or not PCM1 is a direct substrate of Plk4, it is firmly shown that S372 phosphorylation is dependent upon Plk4 kinase activity.").

Minor points:

The way quantifications were done is not entirely clear to me. If I understand it correctly, cells were visualized by eye and categorized. It should be briefly mentioned how it was done.

Yes, we categorised localisation patterns visually. We described how we categorise the patterns in **Supplementary Materials and methods** (pages 5 and 6).

"S372 is conserved within vertebrate Plk4 homologues and locates in the coiled coil rich N-terminal region, although S372 itself is not included in the coiled-coil domain (Fig 2D)" (p.6/7).... Should be: ... vertebrate PCM1...

Thank you very much. This has been corrected.

"Interestingly, the mobility of Plk4-S372A* detected by either anti-GFP or anti-PCM antibodies was slightly faster than that of Plk4-WT*, presumably reflecting the in vivo phosphorylation of Plk4 at this site (Fig 2E)" (p 7) should be corrected to PCM1 where obviously Plk4 is wrong

Thank you for spotting our errors. This has also been corrected.

This sound like a paradox to me: "However, the difference of PCM1 mobility was hardly distinguishable between wild type and kinase-dead Plk4 containing cells.

Collectively, the centriolar satellite component PCM1 is phosphorylated and interacts with Plk4."

To prevent confusion, we removed "However, the difference of PCM1 mobility was hardly distinguishable between wild type and kinase-dead Plk4 containing cells."

Fig. 3B: very few microtubules are seen on the image of the S372 variant tracking. Is that significant, i.e. could the phosphorylation also affect microtubule binding?

As far as we see from microtubule staining, we did not detect substantial alterations of microtubule organisation. The previous image shown in Figure 3B was created from a single projection (not multiply projected or deconvolved), leading to the apparent reduction of microtubule staining. We have removed this image.

The legend of the model now tells us: Whether Plk4 directly phosphorylates PCM1 and where this phosphorylation occurs are currently unknown.

The legend of the model was rephrased for clarification.

Referee #3

We acknowledge his/her careful, useful comments.

1) The phenotypes with respect to satellites are assayed by depleting Plk4 by RNAi. Does the depletion of Plk4 result in loss of centrioles? It should, by published accounts. If so, then the dispersal of satellites is likely to be due to lack of a focused microtubule aster, or possibly lack of centrosomally-anchored microtubules (to the sub-distal appendages, for example). These would be much better experiments if done with centrinone, the new inhibitor of Plk4 described in a paper from Oegema and colleagues. This would allow acute inhibition of Plk4, without loss of centrioles, which requires passage through mitosis, and thus assessment of a Plk4 function in satellites without the confounding effect of loss of centrioles.

As described earlier (to the editor and referee 1's comment), to distinguish the well established centriole duplication defects from the centriolar satellite dispersion phenotype upon Plk4 depletion, we first arrested cells in G1 and then treated cells with Plk4 siRNA, in which Plk4 depletion did/could not lead to the centriole duplication defects. Under this condition, satellite dispersion still occurred (**Figure 1F and G**).

The paper from Oegema and colleagues were published during preparation of our previous manuscript. It would be of great interest to use the Plk4 inhibitor (centrinone) in our experimental system but we believe that this would be the next direction.

2) The putative interaction of Plk4 with PCM1. Is this occurring in the context of the satellite structure, or with the two proteins independently of satellites? The lack of other satellite proteins in the Plk4 IP suggests the latter. What does that mean then? The authors cite another paper (ref. 25) as also identifying a Plk4-PCM1 interaction, but that paper seems to only be about proximity biotin labeling, which says something about the proteins being in the same vicinity. Helpful, but not evidence for a physical interaction.

As stated earlier (to the editor and referee 1's comment), a direct physical interaction between Plk4 and PCM1 using bacterial proteins is now shown (**Figure 2C**).

3) PCM1 phosphorylation. PCM1 has many S/T sites that have been identified as phosphosites in large-scale efforts (see http://www.hprd.org/ptms?hprd_id=02624&isoform_id=02624_2&isoform_name=Isoform_1). It would be helpful to know whether S372 is actually phosphorylated in vivo or not, and whether this depends on Plk4. It could be that the reason the migration of the protein does not appreciably change in the Plk4 kinase dead is that there are many other phosphorylation sites that are unaffected.

Thank you very much for pointing out these two previously published papers, which reported S372 phosphorylation in vivo. We cited these two papers in the text (page 6, "It is of note that previous phosphoproteomic studies also identified S372 as one of the in vivo phosphorylation sites of PCM1 [33, 34]").

Also, as described earlier (to the editor and referees 1's and 2's comments), we prepared an S372-phospho specific antibody and showed that S372 is indeed phosphorylated in the cell.

2nd Editorial Decision

09 November 2015

Thank you for the submission of your revised manuscript to our journal. I apologize for the delay in getting back to you, I was not in the office last week. We have now received the comments from two of the former referees for this manuscript, and as you will see, both support its publication now, pending minor revisions. I would therefore like to ask you to address all concerns and send us a final version of the manuscript as soon as possible. I have asked referee 2 whether or not he agrees with referee 1 that centrinone should be used to inhibit Plk4, and I will let you know as soon as I have heard back from referee 2 whether this experiment should be performed.

Please also check carefully that all information regarding statistics is included in the figure legends,

e.g. the test used to calculate p values needs to be specified in the figure legends. Please also send us a completed author checklist, which you can download from our author guidelines (<http://embor.embopress.org/authorguide#revision>). Please insert page numbers in the checklist to indicate where in the manuscript the requested information can be found.

EMBO press papers do not have supplementary information anymore. Additional figures are called expanded view figures now: EV1, 2, etc. Can you please make these changes in the manuscript text and figures, upload all EV figures as single files, and add the legends for EV figures to the end of the manuscript text? Please also include all materials and methods in the main manuscript file, they are not included in the character count, and we have recently decided that all methods need to be in the main paper.

EMBO reports papers are accompanied online by A) a short (1-2 sentences) summary of the findings and their significance, B) 2-3 bullet points highlighting key results and C) a synopsis image that is exactly 550x200-400 pixels large (the height is variable). You can either show a model or key data in the synopsis image. Please note that text needs to be readable at the final image size. Please send us this information along with the revised manuscript.

I am looking forward to receiving the final manuscript as soon as possible. Please let me know if you have any questions.

REFEREE REPORTS

Referee #1:

In this revised version of the manuscript the authors have included much new data that bears on the common concerns of the reviewers. These concerns focused on two weak points of the original manuscript: whether the satellite dispersal observed upon Plk4 depletion might simply reflect the absence of centrioles, and whether the putative phosphorylation is likely to result from a direct interaction between Plk4 and Pcm1 (and possibly also be a direct phosphorylation by Plk4). These new data address most of serious issues with the manuscript. I still believe, as stated in the original review, that these experiments should be confirmed by treatment with centrinone, a new and potent Plk4 inhibitor. The authors have chosen not to do this, and, if the other reviewers agree, then I think it is acceptable to publish the paper.

Referee #2:

In the newly assembled version of their manuscript T.Toda and colleagues address major concerns that I had raised when reviewing the original submission. In particular, they show that satellite proteins other than PCM1 itself are affected by PCM1's phosphorylation status on S273. These data made it to the suppl. Figures (fair enough), nevertheless, they ought to be quantified to match the quality of the other data in the manuscript. I also appreciate that the authors show phosphorylation dependent interaction of PCM1 with hMsd1, BBS4, Cep290.

Most importantly, however, Hori et al. have generated a reasonably characterized phospho-specific antibody, which enables them to specifically detect S273 on PCM1. Using this tool, they show that S273 phosphorylation occurs in G1 of different human cells. Does the antibody also work robustly in IF? It would be beautiful to see the distribution of S273-phosphorylated PCM1 within the cell during the cell cycle. These data, together with the present data on the direct interaction between PCM1 and Plk4, would add enough substance to the story to have it discussed in public and to support publication in EMBO Reports.

One other still issue remaining is about the dynamics of S273 phosphorylation, which should be discussed appropriately: The phosphomimetic variants rescue ciliogenesis to a significant extent upon knockdown of endogenous PCM1 (remarkable experiment!) although satellite dynamics are frozen. I think the situation may be a complicated one: we have endogenous Plk4, some residual endogenous PCM1 and the exogenously expressed variants that work together to allow at least some ciliogenesis. Does the rescue also work in the absence (knockdown) of Plk4? In any case, I would be

careful proposing a simple model, in which Plk4 sets a phosphorylation that is then required for the function of PCM1 in ciliogenesis in general. If that were the case, we would expect quantitative and stable PCM1 phosphorylation in G1, which is not suggested by the mass spec data, right? The satellite trackings also tell us that this is more complicated, i.e. more dynamic.

Minor issue:

p.7:

"Interestingly, PCM1-S372D* and PCM1-S372E*, but neither PCM1-WT* nor PCM1-S372A*, were capable of rescuing the dispersed localisation of PCM1; ~50% of cells showed aggregated appearance in a similar manner to those in the presence of Plk4 (Fig3C and D)".

I found this sentence difficult to read; what about: "While neither PCM1-WT nor PCM1-S372A* were capable of rescuing the dispersed localization of PCM1, ~50% of cells expressing PCM1-S372D* or PCM1-S372E* showed aggregated appearance in a similar manner to those seen in the presence of Plk4 (Fig3C and D)".

2nd Revision - authors' response

11 November 2015

To the Editor

Please also check carefully that all information regarding statistics is included in the figure legends, e.g. the test used to calculate p values needs to be specified in the figure legends. Please also send us a completed author checklist, which you can download from our author guidelines (<http://embor.embopress.org/authorguide#revision>). Please insert page numbers in the checklist to indicate where in the manuscript the requested information can be found.

We provided all statistic information requested in the corresponding figure legends (and Materials and Methods). For summary, please refer to the checklist attached.

EMBO press papers do not have supplementary information anymore. Additional figures are called expanded view figures now: EV1, 2, etc. Can you please make these changes in the manuscript text and figures, upload all EV figures as single files, and add the legends for EV figures to the end of the manuscript text? Please also include all materials and methods in the main manuscript file, they are not included in the character count, and we have recently decided that all methods need to be in the main paper.

As requested, Supplementary information is converted to a single Expanded view file (Figure legends and Figures EV1-4). All the Materials and Methods are now included in the main text.

EMBO reports papers are accompanied online by A) a short (1-2 sentences) summary of the findings and their significance, B) 2-3 bullet points highlighting key results and C) a synopsis image that is exactly 550x200-400 pixels large (the height is variable). You can either show a model or key data in the synopsis image. Please note that text needs to be readable at the final image size. Please send us this information along with the revised manuscript.

We have provided a short summary, 3 bullet points and a synopsis image.

Referee #1:

In this revised version of the manuscript the authors have included much new data that bears on the common concerns of the reviewers. These concerns focused on two weak points of the original manuscript: whether the satellite dispersal observed upon Plk4 depletion might simply reflect the absence of centrioles, and whether the putative phosphorylation is likely to result from a direct interaction between Plk4 and Pcm1 (and possibly also be a direct phosphorylation by Plk4). These new data address most of serious issues with the manuscript. I still believe, as stated in the original review, that these experiments should be confirmed by treatment with centrinone, a new and potent Plk4 inhibitor. The authors have chosen not to do this, and, if the other reviewers agree, then I think it is acceptable to publish the paper.

We appreciate this referee's support to our work. As for experimentations using centrinone, we are sorry for not following the suggestion, but we believe that this approach belongs to our future direction.

Referee #2:

In the newly assembled version of their manuscript T.Toda and colleagues address major concerns that I had raised when reviewing the original submission. In particular, they show that satellite proteins other than PCM1 itself are affected by PCM1's phosphorylation status on S273. These data made it to the suppl. Figures (fair enough), nevertheless, they ought to be quantified to match the quality of the other data in the manuscript. I also appreciate that the authors show phosphorylation dependent interaction of PCM1 with hMsd1, BBS4, Cep290.

We are glad that this referee in principle is satisfied with our new version. In response to his/her request, we have added quantification data for the dispersion of other satellite components upon Plk4 depletion (the bottom panels of Fig EV3A, B and C).

Most importantly, however, Hori et al. have generated a reasonably characterized phospho-specific antibody, which enables them to specifically detect S273 on PCM1. Using this tool, they show that S273 phosphorylation occurs in G1 of different human cells. Does the antibody also work robustly in IF? It would be beautiful to see the distribution of S273-phosphorylated PCM1 within the cell during the cell cycle. These data, together with the present data on the direct interaction between PCM1 and Plk4, would add enough substance to the story to have it discussed in public and to support publication in EMBO Reports.

Our phospho-specific antibody against phospho-S372 recognised phosphorylated PCM1 only in pull-down samples. Unfortunately, we could not detect phosphor-PCM1 with simple immunoblotting or IF. This is described in the text (page 6) and Materials and Methods (page 14).

One other still issue remaining is about the dynamics of S273 phosphorylation, which should be discussed appropriately: The phosphomimetic variants rescue ciliogenesis to a significant extent upon knockdown of endogenous PCM1 (remarkable experiment!) although satellite dynamics are frozen. I think the situation may be a complicated one: we have endogenous Plk4, some residual endogenous PCM1 and the exogenously expressed variants that work together to allow at least some ciliogenesis. Does the rescue also work in the absence (knockdown) of Plk4? In any case, I would be careful proposing a simple model, in which Plk4 sets a phosphorylation that is than required for the function of PCM1 in ciliogenesis in general. If that were the case, we would expect quantitative and stable PCM1 phosphorylation in G1, which is not suggested by the mass spec data, right? The satellite trackings also tell us that this is more complicated, i.e. more dynamic.

We agree with this referee who raised caution when we interpret the data showing the partial rescue of ciliogenesis defects by producing phosphomimetic PCM1 mutants. In response to this referee's notion, we added the following sentences to the text: page 9, "*Although phosphomimetic PCM1 appeared capable of rescuing ciliogenesis defects, it might need some reservations. For instance, it is possible that phosphomimetic PCM1 might interact with residual wild type PCM1, leading to the formation of functional PCM1 dimers.*"

Regarding the rescue of Plk4- and PCM1- doubly-depleted cells by phosphomimetic PCM1 (PCM1-S372E*), we performed experiments three times and obtained somehow confusing results; one showed **58%** ciliogenesis, while the other two experiments showed **17%** and **8%** ciliogenesis. In all cases, samples containing control empty vectors displayed **15-20%** ciliogenesis. Statistical analysis of these data indicates that the results are "**n.s.**" (please see the figure shown below). As these results show large SD, we feel that we could not draw solid conclusions. It is likely that for proper ciliogenesis to proceed, Plk4 plays an additional role other than phosphorylation of PCM1 or as this referee pointed out and we describe in the text (pages 7 and 9), the dynamic cycle of phosphorylation and dephosphorylation of S372 would be important.

Minor issue:

p.7:

"Interestingly, PCMI-S372D* and PCMI-S372E*, but neither PCMI-WT* nor PCMI-S372A*, were capable of rescuing the dispersed localisation of PCMI; ~50% of cells showed aggregated the presence of Plk4 (Fig3C and D)".

Thank you very much for pointing this out. We have changed sentences as suggested (page 7).

We hope that our responses to you and the reviewers' comments will be sufficient for publication of this manuscript in EMBO Reports.

3rd Editorial Decision

23 November 2015

I have now heard back from referee 2, who agrees with your first suggestion, to discuss the rescue experiments briefly in the manuscript text.

S/he suggests NOT to show the data of the rescue, but to rather add a short discussion on that point, such as „rescue of ciliogenesis after loss of Plk4 and PCMI was not readily observed by expression of PCMI_S273E suggesting that Plk4 may have additional functions in ciliogenesis beyond the phosphorylation of PCMI."

Can you therefore make these changes please? I also noticed that not all figure panels specify the bars and error bars and statistical tests used to calculate p-values. You could also add one sentence at the end of each relevant figure legend with this information, if it is the same for all statistical analyses in the figure. If not, please specify this information for each figure panel.

When uploading your revised manuscript, please upload each figure file separately, and also each expanded view figure. The legends for the expanded view figures need to be added at the end of the main manuscript text.

The manuscript abstract needs to be written in present tense, and we can make these changes for you, if you prefer. I also think that we need to change the manuscript title, can you please suggest a new one?

I look forward to seeing a final, revised version of your manuscript as soon as possible.

3rd Revision - authors' response

24 November 2015

I have now heard back from referee 2, who agrees with your first suggestion, to discuss the rescue experiments briefly in the manuscript text.

S/he suggests NOT to show the data of the rescue, but to rather add a short discussion on that point, such as „rescue of ciliogenesis after loss of Plk4 and PCMI was not readily observed by expression of PCMI_S273E suggesting that Plk4 may have additional functions in ciliogenesis beyond the phosphorylation of PCMI."

As suggested by you and this referee, we added the following sentence in page 9.

"We also produced PCMI-S372E upon double depletion of Plk4 and PCMI; under this condition, however, the rescue of ciliogenesis was not readily observed, suggesting that Plk4 may have additional functions in ciliogenesis beyond the phosphorylation of PCMI."*

I also noticed that not all figure panels specify the bars and error bars and statistical tests used to calculate p-values. You could also add one sentence at the end of each relevant figure legend with this information, if it is the same for all statistical analyses in the figure. If not, please specify this information for each figure panel.

As requested, in all figure legends corresponding to relevant panels, we specified the bars, error bars and statistical tests used to calculate p-values.

When uploading your revised manuscript, please upload each figure file separately, and also each expanded view figure. The legends for the expanded view figures need to be added at the end of the main manuscript text.

We followed your request written above.

The manuscript abstract needs to be written in present tense, and we can make these changes for you, if you prefer. I also think that we need to change the manuscript title, can you please suggest a new one?

May I ask the journal to change tense from past to present on behalf of us?

Regarding the title, we could suggest the following new title,

Previous:

“Plk4 regulates centriolar satellite integrity and promotes ciliogenesis through PCM1 phosphorylation”

New:

“A non-canonical function of Plk4 in centriolar satellite organisation and ciliogenesis through PCM1 phosphorylation”

However, with regards to this issue, we are ready to follow your suggestion if you have.

4th Editorial Decision

30 November 2015

I am very pleased to accept your manuscript for publication in the next available issue of EMBO reports. Thank you for your contribution to our journal.